# Internal Hernia and Volvulus in an Adult Male Caused by Meckel’s Diverticulum: A Case Report

**DOI:** 10.3390/medicina57050443

**Published:** 2021-05-03

**Authors:** Konstantinos Sapalidis, Christina Sevva, Vasiliki Magra, Vasiliki Manaki, Charilaos Koulouris, Panagiota Roulia, Athanasios Katsaounis, Despoina Vasileiou, Stelian Pantea, Isaak Kesisoglou

**Affiliations:** 13rd Surgery Department, AHEPA University Hospital, School of Medicine, Faculty of Health Sciences, Aristotle University of Thessaloniki, 54621 Thessaloniki, Greece; christina.sevva@gmail.com (C.S.); valia.magra@gmail.com (V.M.); vassiamanaki@gmail.com (V.M.); charilaoskoulouris@gmail.com (C.K.); panagiotar96@gmail.com (P.R.); athanasioskatsaounis@gmail.com (A.K.); desva4@yahoo.gr (D.V.); ikesis@hotmail.com (I.K.); 2UMF “Victor Babes”, Surgery Clinic II, 300041 Timisoara, Romania; panteastelian7@gmail.com

**Keywords:** meckel’s diverticulum, ileum, internal hernia, volvulus, case report

## Abstract

*Background:* Meckel’s diverticulum is a common congenital abnormality of the gastrointestinal tract encountered in about 1–3% of the general population. Although most patients remain asymptomatic, a minority will experience serious complications such as acute abdomen, haemorrhage or obstructive ileus. Of all patients presenting with symptoms of obstruction due to Meckel’s diverticulum 7–18% is due to volvulus. *Case Report:* A 39-year-old male with multiple previous episodes of obstructive ileus presented with an acute abdomen. An exploratory laparotomy was performed in order to reveal the cause of the obstruction. An internal hernia with ileal volvulus and a Meckel’s diverticulum was found, which was later confirmed by histopathological examination. *Conclusion:* Meckel’s diverticulum is a rare cause of acute abdomen and obstructive ileus which should be considered when the symptoms date back to childhood. The difficulty of preoperative diagnosis dictates the need for exploratory laparoscopy or laparotomy as diagnostic tools.

## 1. Introduction

Meckel’s diverticulum (MD) is the most common true congenital diverticulum found in the gastrointestinal tract. It is the result of incomplete regression of the omphalomesenteric duct and it contains all intestinal wall layers [1]. Meckel’s diverticulum has a prevalence of 0.3–2.9% in the general population and accounts for approximately 90% of all omphalomesenteric duct malformations. It is usually identified during abdominal operations. However, with modern improved imaging means, epidemiology of the disease should be reconsidered [2]. MD is usually asymptomatic [3], however 4–6% of these patients are at risk of serious complications, most commonly bowel obstruction and diverticulitis in adults and intestinal bleeding in children [4].

We report a rare case of a 39-year-old male presenting with ileus caused by internal herniation and ileal volvulus due to Meckel’s diverticulum. Just a few cases similar to this one have been reported in the existing literature. MD should be considered in such cases as part of the differential diagnosis.

## 2. Case Presentation

A 39-year-old male presented to the Emergency Department complaining of abdominal distention, constipation, nausea, and vomiting. The symptoms started two days before presentation and were gradually intensifying. The patient also mentioned several episodes of obstructive ileus in previous years. Some of these episodes dated back to childhood and many required hospitalisation without previous operative intervention. 

Laboratory results showed leukocytosis with a White Blood Count (WBC) of 15.08 K/μL with 90.9% Neutrophils. C—Reactive Protein (CRP) was also elevated at 29.70 mg/dL. The patient had elevated blood glucose (136 mg/dL) and a mildly raised Creatinine at 1.12 mg/dL. All other laboratory results were within normal range. An abdominal radiograph performed in the Emergency Department showed multiple air-fluid levels and dilated intestinal loops, indicating a case of ileus. An abdominal ultrasound was performed and Computed Tomography (CT) of the abdomen and pelvis was ordered, which initially indicated non-obstructive ileus. A nasogastric tube was inserted and the patient was admitted in the surgical ward for monitoring. The patient initially improved with conservative management—antiemetics and nasogastric tube, which was removed 48 h after admission. His clinical condition deteriorated acutely with worsening abdominal pain and multiple episodes of vomiting. His White Blood Count returned to normal (7.71 K/μL), but Hemoglobin (Hb) levels dropped to 13.2 g/dL from 15.4 g/dL. An emergency exploratory laparotomy took place. After thorough small bowel inspection the cause of the ileus was revealed. A Meckel’s diverticulum was present approximately 30 cm from the ileocaecal valve causing internal herniation and ileal volvulus (Figure 1). Bowel resection of the ischaemic ileum containing Meckel’s diverticulum was performed followed by a side-to-side anastomosis. An abdominal drain was inserted in the pouch of Douglas. The specimen was sent for histopathological examination which revealed that the 18cm-long part of the small intestine that was resected contained a 5.5 × 4.5 × 1 cm^3^ saccular dilatation (Figure 2). Numerous eosinophils, neutrophils, lymphocytes, plasmatocytes and histiocytes were present and the lesion was confirmed as Meckel’s diverticulum.

The patient was transferred to the Intensive Care Unit (ICU) post-operatively. He became pyrexial on day 2 post-operatively. The patient remained consistently pyrexial until day 8 post-operatively. He had thorough biochemical and radiological investigation, including two Computed Tomography (CT) scans as well as blood and drain fluid cultures. Although the CT scans did not reveal a specific focus of infection or intra-abdominal collection, blood cultures were positive for Staphylococcus epidermidis potentially due to hospital acquired infection. Cultures from the drain revealed Enterobacter aerogenes. His treatment consisted of tigecycline, colistin and andulafungin for eight days, which he tolerated well, according to infectious diseases department advice. Both cultures turned negative on the 15th post- operative day. The remainder of his inpatient stay was uneventful until his discharge on day 18 post-operatively. 

The manuscript has been developed according to the CARE (CAse REport) guidelines checklist [5].

## 3. Discussion

Meckel’s diverticulum is the most common congenital abnormality of the gastrointestinal tract. It remains asymptomatic in most individuals and may be found incidentally during other surgical procedures. It can rarely present with acute symptoms and abdominal complications [6]. These defer among different age groups. During childhood, intestinal bleeding is the commonest complication due to presence of ectopic gastric mucosa in the diverticulum [4,7,8]. Although potentially life-threatening, bleeding in the paediatric population is very uncommon compared to complications affecting the adult population. Complications can be categorized into three subgroups: those caused by bowel obstruction, those caused by an inflammatory process and those caused by tumours of the diverticulum [4]. Bowel obstruction may occur due to Meckel’s diverticulitis, enteric volvulus, intussusception, Litre’s Hernia and enterolith formation ileus [4,7,8]. Inflammatory process is the second most common complication and includes Meckel’s diverticulitis and perforation both related to obstruction of the diverticulum or peptic ulceration due to ectopic gastric tissue [4,9,10]. Finally, benign and malignant neoplasms—lipomas, vascular hamartomas, neuroendocrine, carcinoid and mesenchymal tumours, adenocarcinomas—are less frequently accountable for complications [4,7,11].

This patient presented with symptoms of ileus. However, the presence of volvulus due to Meckel’s diverticulum was not identified until the patient underwent exploratory laparotomy. Volvulus is uncommon and—according to the literature—responsible for 7–18% of patients with intestinal obstruction caused by Meckel’s diverticulum [12].

Preoperative diagnosis of Meckel’s diverticulum can be challenging. Screening the general population is neither cost effective nor practical due to the rarity of the condition and especially its complications. It also carries significant morbidity as the options for such screening would include CT scanning or diagnostic laparoscopy, as ultrasound has low diagnostic sensitivity and Magnetic Resonance Imaging (MRI) is not easily available and very expensive. The diagnostic process involves radiological imaging (X-rays, computed tomography, magnetic resonance imaging, ultrasonography, barium examination), angiography, nuclear imaging (Technetium-99m pertechnetate scanning), capsule-endoscopy and double-balloon enteroscopy, CT and Magnetic Resonance (MR) enterography [4,6,7,13]. Although availability of at least some of these diagnostic means is feasible, their specificity and sensitivity are low [2,13]. Diagnostic laparoscopy or laparotomy are more accurate methods of diagnosing Meckel’s diverticulum but are not first line investigations [6,14]. Most commonly, Meckel’s diverticulum is identified incidentally during another surgical procedure [4,6].

Management of Meckel’s diverticulum relies on patient symptomatology. In acute clinical presentations, most authors agree that an exploratory laparotomy is indicated in order to identify and treat the cause [13]. In case resection is required, it should include part of the ileum as well as Meckel’s diverticulum via diverticulectomy or wedge and segmental resection, depending on the size of the diverticulum [6,13,15]. In asymptomatic patients with incidentally discovered Meckel’s diverticulum the unanimous opinion on surgical management is yet to be reached. In recent years many authors tend to have the notion that specific risk factors must be present in order to proceed to a diverticulectomy or resection [4,6,15]. Those risk factors include patient age < 50 years, male gender, diverticulum length > 2 cm and presence of ectopic or abnormal tissue within a diverticulum [15].

In our case, exploratory laparotomy was performed in order to identify and treat the cause of the obstruction after failure of conservative management. The patient also met three of Schwenter’s risk factors, which support the indication for emergency surgical intervention. Ref [16] The pathological lesion was identified intraoperatively and a resection of the ileum including the diverticulum was performed.

## 4. Conclusions

Meckel’s diverticulum is a rare cause of acute abdomen. All patients with complications from Meckel’s diverticulum should be managed operatively. Due to the difficulties in preoperative diagnosis physicians should have high suspicion for this condition when there is diagnostic uncertainty and the patient’s history, age group and clinical picture indicate so.

Due to the diagnostic challenges mentioned and the impracticality of screening the general population, it is suggested that the diagnostic process includes early diagnostic laparoscopy in young patients presenting with obstructive ileus failing conservative management.

## Figures and Tables

**Figure 1 medicina-57-00443-f001:**
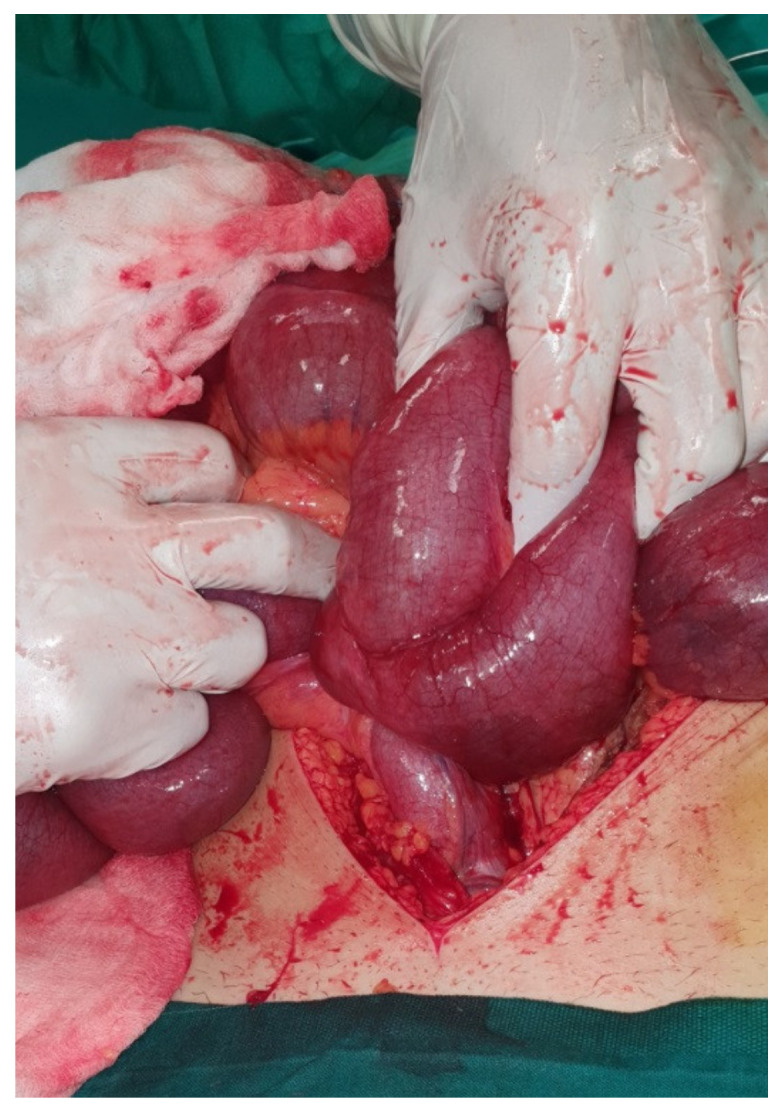
Intraoperative image of the volvulus and internally herniated loop of ileum.

**Figure 2 medicina-57-00443-f002:**
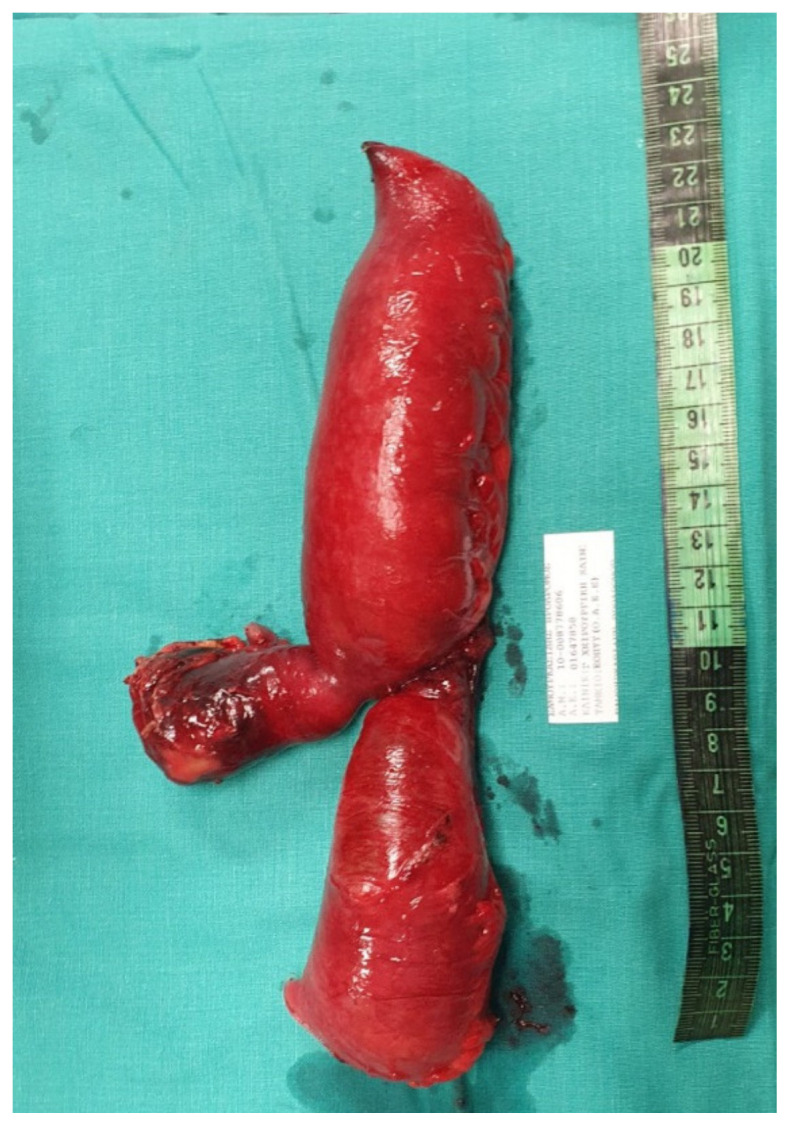
Post-resection image of the specimen of loop of ileum, including Meckel’s diverticulum.

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
