# Peer review of "Internal Hernia and Volvulus in an Adult Male Caused by Meckel’s Diverticulum: A Case Report"

_medicina, 2021, doi:10.3390/medicina57050443_

Round 1

Reviewer 1 Report

The paper presents a case of bowel obstruction secondary to volvulus due to the Meckel diverticulum. The case presentation is clearly described and with rich imagistic documentation. However, although it presents a rare pathological encounter in surgical practice, the article does not bring any novelty, nor as diagnostic or management of this pathology.

Major point:

- the postoperative evolution should be better discussed: the possible etiology (endogenous/ hospital-acquired infection) and specific treatment

Author Response

Thank you for your useful comments. We made the appropriate revisions as you suggested. We further discussed the post-operative management of the patient and we clarified the exact purpose of this report and our suggestions concerning the diagnostic process of such cases. 

Reviewer 2 Report

This is an interesting case report of acute abdomen caused by a Meckel's diverticulum in an adult aged 39 years. I have some minor suggestions for revision:

  1. Please ensure that your paper adheres to the CARE guidelines for the writing of case reports.
  2. The introduction is too short. 
  3. Is screening for Meckel's diverticulum feasible?
  4.  It would be interesting to see some pathology slides as well.
  5. The authors should also design an algorithm regarding the treatment/management of this condition to improve the novelty of the paper.
  6. Please revise the references, they are not in the MDPI style. Also, ref. 14 and 15 are the same.

Author Response

Thank you for your useful comments. We made the appropriate revisions as you suggested.

  1. Regarding CARE guidelines for case reports, follow up is scheduled in 6 months so it was not possible to mention it.
  2. We extended the introduction and clarified the aim of this report.
  3. Screening is not cost effective due to the rarity of the condition and its complications. The available diagnostic means are either expensive or carry high morbidity.
  4. Unfortunately, after contacting the pathology lab, slides are not available
  5. We suggested a more extensive diagnostic process emphasizing on the need of early diagnostic laparoscopy in young individuals with obstructive ileus
  6. We revised the references.

Round 2

Reviewer 1 Report

All the comments in the first round of review were addressed in the revised version. Although the level of novelty is low, the paper is well written and the case is well documented

Author Response

Thank you very much for your useful comments. In my 30 year experience in surgery department in the central university hospital in Thessaloniki I have only seen complications by Meckel's diverticulum three time. As mentioned in the manuscript , it is suggested that the diagnostic process should include early diagnostic laparoscopy in young patients presenting with obstructive ileus failing conservative management. Due to the difficulties in preoperative diagnosis physicians should have high suspicion for this condition when there is diagnostic uncertainty and the patient’s history, age group and clinical picture indicate so.